# Environmental Effects of Sport Horse Production Farms in Argentina

Mariana M. Vaccaro [1,2], Alejandra V. Volpedo [2,3,*], Alberto Garcia-Liñeiro [1] and Alicia Fernández-Cirelli [2,3]

1   Cátedra de Salud y Producción Equina, Facultad de Ciencias Veterinarias, Universidad de Buenos Aires, Buenos Aires C1427CWO, Argentina; mvaccaro@fvet.uba.ar (M.M.V.); garcialineiro@fvet.uba.ar (A.G.-L.)
2   Instituto de Investigaciones en Producción Animal (INPA) Buenos Aires, CONICET-Universidad de Buenos Aires, Argentina. Av. Chorroarín 280, Buenos Aires C1427CWO, Argentina; afcirelli@fvet.uba.ar
3   Centro de Estudios Transdisciplinarios del Agua (CETA), Facultad de Ciencias Veterinarias, Universidad de Buenos Aires, Buenos Aires C1427CWO, Argentina
*   Correspondence: avolpedo@fvet.uba.ar

**Abstract:** Argentina is one of the countries that exports animals for equestrian sports. This paper analyzes the environmental effects of sport horse production farms in Argentina and proposes actions to minimize the environmental effects of this type of production. Twenty-six sport horse production farms in the province of Buenos Aires were studied. The proximity of the farms to a surface water body, the destination of the stall bedding, management practices and whether they receive veterinary advice were the characteristics analyzed in relation to feed, its composition and water consumption according to the performance of the animals. A nominal qualitative analysis on the impact was carried out considering three impact categories: low, medium and high. The association between the four environmental variables analyzed has shown that only two farms have a low environmental impact, while eighteen farms have a medium impact and five farms have a high impact. The results show that the role of the professional veterinarian is key in minimizing environmental impact and that the management of excretions and stall bedding should be reviewed in order to reduce the impact. This paper presents recommendations associated with water use, feed and manure management.

**Keywords:** water; feed; pollution; sport equine production

## 1. Introduction

The environmental effects of livestock productions have been studied worldwide in recent decades [1–4]. However, these are focused on cattle [5–7] and there are very few studies on horses and more specifically on sport horses [8–13].

On the other hand, considering the hours that sport horses spend in the stalls, the abundant droppings they produce, the lack of environmental protocols for the destination of the stall bedding and the amount of water required for direct and indirect consumption, among other factors, we can demonstrate that this production generates relevant environmental effects that must be minimized [10,12]. For example, the concentration of nutrients in the diet and overfeeding is one of the main dilemmas in relation to the conformation of manure and its environmental implications [9,14,15].

Poor manure management practices could result in increased movement of sediments, nutrients and xenobiotics into surface or groundwater, altering their quality [14]. The management of manure and bedding is the subject of much research worldwide [14,15] and is scarce in Argentina [10] given the potential effect that they can cause on the environment. The breeds of sport horses that are bred in Argentina are American trotter (AT), Polo Argentino (PA), Silla Argentino (SA) and Zangershaide (Z). Argentina is one of the countries that exports animals for equestrian sports. In the last 5 years, our team has increased the study of different factors associated with the environmental effects of the production of sport horses; analyzing the quality of drinking water, determining the presence of trace

elements in food, excreta and beds of the boxes as well as quantifying the water used for consumption, cleaning and cleaning of the boxes [10–12]. Considering these previous works, and performing a comprehensive analysis, this paper analyzes the environmental effects of sport horse production farms in Argentina and proposes actions to minimize the environmental effects of this type of production.

## 2. Materials and Methods

Twenty-six sport horse production farms in the province of Buenos Aires were studied (Figure 1, Table 1). The proximity of the farms to a surface water body, destination of the stall bedding, management practices and whether they receive veterinary advice were the characteristics analyzed in relation to feed, its composition and water consumption according to the performance of the animals. The breeds of sport horses studied are American trotter (AT), Polo Argentino (PA), Silla Argentino (SA) and Zangershaide (Z).

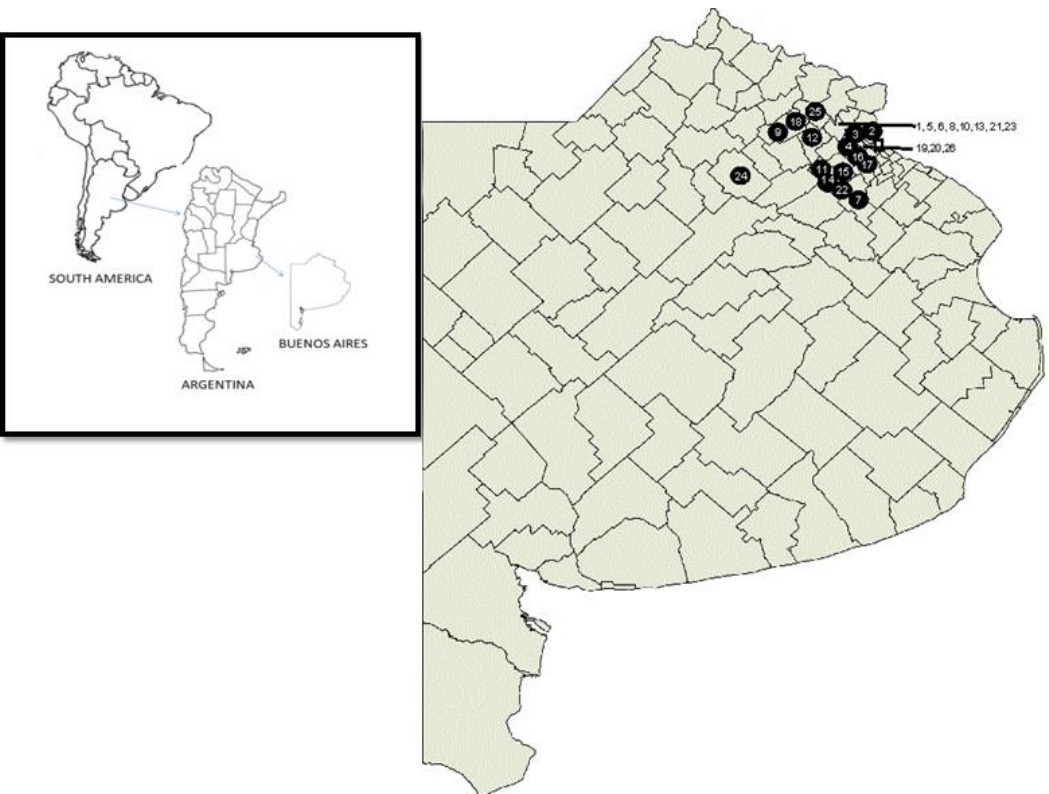

**Figure 1.** Geographic distribution of sport horse production farms. Sampling sites: 11-14-15-22—General Las Heras, 2—San Isidro, 3—Malvinas Argentinas, 4-16-17—Moreno, 1-5-6-8-10-13-21-23—Pilar, 7—Cañuelas, 9-18—San Andrés de Giles, 12—Open Door, 19-20-26—Hurlingham, 24—Chivilcoy, 25—Exaltación de la Cruz. Square: city of Buenos Aires.

This information was obtained through a survey that was administered to the producers and also with interviews of the veterinary professionals of the farms. The characterization was based in relation to the number of animals, performance and water consumption [12].

A nominal qualitative analysis on the impact was carried out considering three impact categories: low, medium and high. A normalization table of the studied variables was prepared (Table 2), and values of 5 to 24 points were assigned to low impact, values from 25 to 50 points to medium impact and values > 50 points to high impact.

**Table 1.** Sport horse production farms' characterization synthesis.

| N | Quantity | Breed | Feed | Supplement | Time in the Stall (h) | Bedding Type |
|---|---|---|---|---|---|---|
| 1 | 24 | SA | Ab, B | Oat | 12 | Industrial wood shavings |
| 2 | 200 | PA | Abp, B | Corn, oat, barley, growth promoters | 23 | Straw–wood shavings |
| 3 | 40 | SA | Abp | Corn, oat, barley, electrolytes, oils | 14 | Wood shavings |
| 4 | 17 | SA | Abp | Corn, oat, barley, electrolytes, oils, vitamins | 16 | Straw–wood shavings |
| 5 | 50 | SA | Abp | Oat, oils, vitamins | 10 | Wood shavings |
| 6 | 40 | SA | Abp | Corn, oat, barley, vitamins | 10 | Wood shavings |
| 7 | 10 | SA | Ab | Oat, oils, electrolytes, vitamins | 10 | Wood shavings |
| 8 | 62 | SA | Abp | Oils, electrolytes, vitamins | 12 | Wood shavings |
| 9 | 48 | PA | Ab | Oat | 12 | Straw |
| 10 | 11 | SA | Abp | Oat, oils, electrolytes, vitamins | 14 | Wood shavings |
| 11 | 17 | SA | Abp | Oat, oils, electrolytes, vitamins | 15 | Wood shavings |
| 12 | 18 | SA | Ab | Barley | 14 | Wood shavings |
| 13 | 16 | SA | Ab | No supplement | 18 | Wood shavings |
| 14 | 280 | Z | Ab | Oat, oils, vitamins | 12 | Wood shavings |
| 15 | 21 | SA | Abp | Oat, barley, oils, electrolytes, vitamins | 14 | Wood shavings |
| 16 | 40 | SA | Ab | Oat, electrolytes, vitamins | 18 | Straw–wood shavings |
| 17 | 22 | SA | Abp | Electrolytes, vitamins | 16 | Straw–wood shavings |
| 18 | 410 | PA | Ab | Oat | 12 | Wood shavings |
| 19 | 14 | AT | Ab, B | Oat, barley, oils | 22 | Wood shavings |
| 20 | 30 | AT | Abp | Corn, growth promoters | 22 | Wood shavings |
| 21 | 46 | SA | Ab, B | Oat, oils, electrolytes, vitamins | 10 | Industrial wood shavings |
| 22 | 30 | Z | Ab | Oat, oils, vitamins | 23 | Wood shavings |
| 23 | 30 | SA | Abp | No supplement | 12 | Wood shavings |
| 24 | 21 | SA | Abp | Oat, electrolytes, vitamins, growth promoters | 10 | Wood shavings |
| 25 | 14 | SA | Ab | Oat | 14 | Wood shavings |
| 26 | 15 | SA | Abp | Oat | 18 | Straw–wood shavings |

Feed: Ab—alfalfa (bundles), Abp—alfalfa (bundles and pellets); B—balance; N: sport horse production farms; breeds of sport horses: AT—American trotter, PA—Polo Argentino, SA—Silla Argentino, Z—Zangershaide.

**Table 2.** Standardization of variables related to environmental effect.

| Assessment | 1 | 5 | 10 |
|---|---|---|---|
| Veterinary attention frequency | veterinarian staff providing advice on a daily basis | Veterinarian staff providing advice three times a week | Veterinarian staff providing advice once a week |
| Balanced diet prescribed by a professional veterinarian, nutritionists and animal technicians | Balanced diet prescribed exclusively by a professional veterinarian, nutritionists and animal technicians | Diet supplemented by non-professional staff with partial advice from a professional veterinarian, nutritionists and animal technicians | Non-balanced diet administered by non-professional staff |
| Trough usage | Concrete trough | Permeable trough, feed is on plastic or tarpaulin on the ground | No trough is used, feed is on the ground |
| Destination and elimination of wastes (stall bedding and animal excretions) | Bedding and excretions are left in a place with floor and roof | Bedding and excretions are left in a place with floor but no roof. Rainfall is controlled so as to bear in mind moments of no rainfall. | Bedding and excretions are left in a place with neither floor nor roof and rainfalls are not controlled. |
| Proximity to surface water body | >1000 m | 300–1000 m | <300 m |

The variables studied had the following characteristics (Table 2):

- The value of 1 was given when the farm is located at a distance greater than 1000 m from a surface water body. This same value was considered in the case that stall bedding has a final destination in a specific area; with waterproofed ground and

under a roof to avoid leaching; and when there is strict veterinary control on the amount and type of feed given to the animals.

- The value of 5 was given when the farm is at a distance between 300–1000 m from a surface water body. A value of 5 was also given when the stall bedding has a final destination in the open air, but there is some type of management on it, such as covering or resting on compacted ground or on a waterproof surface, etc., and when there is some type of veterinary control on the amount and type of feed given to the animals.

- The value of 10 was given when the farm is close to a surface water body (<300 m); likewise, this value corresponded to the farms that have stall bedding in the open air and on the ground. A lack of a strict veterinary control on the amount and type of feed given to the animals was also rated with 10 points.

The assessment values 1, 5 and 10 used to standardize the variables are based on the exclusive knowledge of the veterinary professionals. For example, the composition of a balanced diet must be prescribed exclusively by veterinarians. This is essential since many authors such as those of [16] state that overfeeding is a health issue for sport equines and also has an important environmental impact [9,14,15]. The supplementation with inorganic phosphorus must be controlled in horses' diets to diminish its excretion in feces, since this is the main way of elimination and, considering the rise of equine production globally, there might be an increase in the environmental risks produced by phosphorus leaching [17]. Associated with this, the presence of feed in troughs is a relevant strategy for reducing the environmental impact, which has also been observed by researchers.

The transportation and elimination of wastes (stall bedding and excretions) is essential in the production of sport horses, and producers generally do not manage them properly. This type of waste is generated in significant volumes, and is scarcely documented [18].

Even though the proximity to surface water bodies is a factor that cannot be modified, since it is associated with the location of the farm, there are studies that report that the proximity of this type of production to water bodies generates an impact due to the issue of runoff which provides nutrients to these bodies of water.

A proximity <300 m is considered close [5], and studies on effluents from intensive cattle, swine and poultry systems [19] in Argentina could be taken as reference (Table 2). With this information, a matrix was constructed and the farms with low, medium or high environmental impact were identified.

## 3. Results and Discussion

A total of 46.15% of the farms studied are located near bodies of water at different distances (Table 3). It is observed that 7.69% of the farms have water bodies nearby (<300 m), while 38.46% have them at a distance between 300 and 1000 m. The farms that are more than 1000 m away are considered to be not close to surface water bodies.

**Table 3.** Sport horse production farms located different distances from water bodies.

| N | Location | Distance from Water Body |
|---|---|---|
| 1 | Pilar | Luján River 2 km, stream 300 m |
| 2 | San Isidro | Artificial lagoon 500 m |
| 3 | Malvinas Argentinas | Basualdo stream 1 km |
| 6 | Pilar | Artificial lagoon 100 m |
| 14 | Gral. Las Heras | Las Heras stream 200 m |
| 19 | Hurlingham | Reconquista River 500 m |
| 20 | Hurlingham | Reconquista River 1000 m |
| 21 | Pilar | Luján River 300 m |
| 22 | Gral. Las Heras | Stream 1000 m |
| 23 | Pilar | Luján River 1.5 km, Pinazo stream 300 m |
| 24 | Chivilcoy | Chivilcoy stream 700 m |
| 26 | Hurlingham | Reconquista River 1 km |

N: Sport horse production establishments.

The water bodies associated with the studied equine production farms are lotic systems in the suburban area of Buenos Aires associated with the Luján River and the Reconquista River. The Luján River basin has a length of 128 km that begins at the confluence of the Durazno and Los Leones streams. Farms 1, 23 and 21 in the Pilar area are close to this basin (Figure 1).

The Reconquista River basin has an approximate extension of 1738 km². The basin comprises 134 water courses that run 606 km, of which 82 correspond to the Reconquista River. The Reconquista River presents general characteristics typical of a plain course; it is affected by the rainfall regime, the fluctuations of the Paraná River, the tides of the Río de la Plata and the southeast winds. Farms 3, 14, 19, 20, 26 and 22 in the districts of Malvinas Argentinas, Hurlingham and Gral. Las Heras are related to this basin (Figure 1).

The Chivilcoy stream is largely tubed and is constituted by the stormwater drainage of the city of Chivilcoy flowing into the Salado River. Farm 24 is linked to this stream.

All the farms studied eliminate bedding and excretion waste in open-air deposits intended for that purpose directly onto the ground. This situation is environmentally complex, since excretions may contain different compounds.

The discarded bedding can increase the volume of manure by two or three times, depending on the type of bedding used [20]. Additionally, these authors estimate that an adult horse kept in a stall requires 5 to 10 kg of bedding per day, which must be changed regularly, considering that it produces 25 kg of manure per day—around 10 tons per year—which would release 4.5 kg of N, 1.7 kg of $P_2O_5$ and 2.4 kg of $K_2O$ per ton into the environment.

The association between the four environmental variables analyzed is shown in Table 4.

**Table 4.** Characterization of the impacts of each sport horse production farms in relation to the analyzed variables and its assessment.

| N | VF | BD | Trough Usage | W | SW | Scoring | IC |
|---|---|---|---|---|---|---|---|
| 1 | 5 | 5 | 1 | 10 | 1 | 22 | L |
| 2 | 5 | 5 | 5 | 10 | 5 | 30 | M |
| 3 | 5 | 5 | 1 | 10 | 5 | 26 | M |
| 4 | 5 | 5 | 1 | 10 | 1 | 22 | L |
| 5 | 5 | 5 | 1 | 10 | 1 | 22 | L |
| 6 | 5 | 5 | 1 | 10 | 10 | 31 | M |
| 7 | 5 | 5 | 1 | 10 | 5 | 26 | M |
| 8 | 1 | 5 | 1 | 10 | 1 | 18 | L |
| 9 | 10 | 5 | 1 | 10 | 1 | 27 | M |
| 10 | 5 | 5 | 1 | 10 | 1 | 22 | L |
| 11 | 10 | 5 | 1 | 10 | 1 | 27 | M |
| 12 | 1 | 5 | 1 | 10 | 1 | 18 | L |
| 13 | 5 | 5 | 1 | 10 | 1 | 22 | L |
| 14 | 10 | 5 | 1 | 10 | 1 | 27 | M |
| 15 | 10 | 5 | 1 | 10 | 1 | 27 | M |
| 16 | 5 | 5 | 5 | 10 | 1 | 26 | M |
| 17 | 5 | 5 | 1 | 10 | 1 | 22 | L |
| 18 | 10 | 5 | 1 | 10 | 1 | 27 | M |
| 19 | 10 | 5 | 1 | 10 | 5 | 31 | M |
| 20 | 10 | 5 | 1 | 10 | 5 | 31 | M |
| 21 | 5 | 5 | 1 | 10 | 10 | 31 | M |
| 22 | 10 | 5 | 1 | 10 | 5 | 31 | M |
| 23 | 5 | 5 | 1 | 10 | 5 | 26 | M |
| 24 | 5 | 5 | 1 | 10 | 5 | 26 | M |
| 25 | 5 | 5 | 1 | 10 | 1 | 22 | L |
| 26 | 5 | 5 | 5 | 10 | 5 | 30 | M |

N: sport horse production establishments, BD: balanced diet prescribed by professional veterinarian, IC: impact category, L: low impact, M: medium impact, SW: proximity to surface water body, VF: veterinary attention frequency, W: destination and elimination of wastes (stall bedding and animal's excretions).

In this sense, it can be observed that eight farms (1, 4, 5, 8, 10, 12, 13 and 25) have a low environmental impact, while the rest have a medium impact (Table 4). Looking at the variables, it is evident that farms with a high environmental impact are those that have a lower frequency of professional veterinary advice, do not have proper management of bedding and excretions and are also located near water bodies (Table 3). Although the latter variable cannot be modified, the first two can, so improving the management of bedding and excretions, as well as the frequency of professional veterinary presence, could improve the sustainability of the farm.

Something similar happens in farms whose impact is medium, as the only thing that really sets them apart from farms with high impact is their proximity to water bodies.

Westendorf et al. [9] surveyed horse breeders in New Jersey and found that 60% of them balanced diets on their own or had no feeding plan. These results show that the role of the professional veterinarian is key in minimizing environmental impact and that the management of excretions and stall bedding should be reviewed in order to reduce their impact, for example, by seeking sustainable alternatives for their reuse.

### 3.1. Recommendations

The increasing levels of international equestrian sports and the socioeconomic benefits derived from them constitute a good reason to develop standards and terms relevant to a distinguished subpopulation of competition horses. Some of these proposed standards are specifically associated with water use, feed and manure management.

### 3.1.1. Water Use

The results show that the production farms in the suburban areas of Buenos Aires studied generally have good water quality for production, although there may be different problems to consider, such as excessive salinity and the presence of nitrates of anthropogenic origin, especially in farms close to the Autonomous City of Buenos Aires.

Regarding the use of water, it must be considered that animals' grooming represents the highest percentage of water used and consumed, after the amount of water indirectly associated with feed (forage and supplements), with a smaller but considerable quantity of water used for cleaning the stalls.

The type of animals and the activity they do determine the amount of water used and consumed, since animals that develop more activity consume more water. For this reason, the calculations of the water footprint of this production must consider—in addition to the water for direct consumption by animal drinking and the indirect water present in the feed—the use of water for cleaning the animals as well as the farms in order to avoid underestimating the volume of this resource that the agricultural production needs.

### 3.1.2. Feeding and Manure Management

With regard to what the animals consume and, therefore, eliminate, it is evident that the way in which bedding waste and excretions are placed outside the stalls is a key factor. If there is rainfall, nutrients and elements present in bedding waste and excretions can leach and reach the underground layers or surface water bodies nearby through runoff, since the bedding waste disposed can accumulate, if the time between elimination and removal is not considered. Manure and bedding waste can be generated in large volumes when they are not removed daily.

Formulating equine rations should preferably be based on nutritional analysis. The variability highlights the importance of forage analysis when designing rations based on it, especially in high-performance sport horses [21]. The organic phosphorus (P) in feed can meet the needs of sport horses, and inorganic phosphorus supplementation is not needed [22]. The fact that these farms are fully advised by veterinarians would guarantee early detection of possible disease symptoms and the correct choice of ingredients in their diets, but periodic visits by professionals should be guaranteed. For example, diets that contain only forage cut at an early stage of plant maturity can satisfy the high energy

requirements of horses in training by supplementing only with minerals and vitamins, with an understanding that, in this way, overfeeding is reduced [21]. Following the studies of Connysson et al. [23], nutritional analysis is recommended and an estimate of the energy content should be made.

Forage legumes tend to have higher protein and calcium content than forage grasses, and this should be taken into consideration [24].

On the other hand, we must also consider other factors such as forage quality, foreign bodies and feed palatability. These factors can have a significant impact on the overall management and well-being of sport horses. Forage quality plays a crucial role in meeting the nutritional requirements of horses and can affect their performance and health. It is important to assess the nutritional composition, digestibility and potential contaminants in the forage [13]. Furthermore, feed palatability is a critical aspect for ensuring that horses consume an adequate and balanced diet. Poor feed palatability can lead to reduced intake and potential nutrient deficiencies. This was demonstrated, for example, by Vinassa et al. [25] in ponies where palatability was studied in relation to temperamental characteristics and their lateralization response.

In a balanced equine diet, mineral concentrations are normally formulated to meet the requirements of a horse working lightly [26]. Another author determined that, when they are fed with medium-quality forage, the requirements for the same category are covered [15]; in both cases, it is complemented with concentrated feeds and, in this sense, both authors agree with Uotila et al. [27] who believe that, when supplementing the diet, the true concentrations of minerals in the other components of the intake are not considered. The study conducted by Harper et al. [28] can also be mentioned, which demonstrated that the intake of crude protein exceeded the NRC recommendations by an average of 50%. In Argentina, there is no register of studies with these characteristics, but it can be mentioned that a high protein diet results in high nitrogen, both in urine and feces. The increase in the rate of nitrogen excretion can lead to acidification and eutrophication of the environment.

Regarding to the nutritional requirements of horses in Argentina and in the world, this topic still requires further research. Conducting more studies on equine nutrition would allow this industry to achieve higher productivity through a well-balanced economic formulation.

It could be concluded that supplying balanced diets and not using unnecessary dietary supplements favors the production of sport horses since money is saved and the environment is protected. The survey designed for the present investigation has been a good opportunity to evaluate some of the management practices, allowing to identify the need to carry out further activities for equine producers in order to promote sustainability of sport horse production farms.

On the other hand, each sport horse production farm should implement an operational plan that includes comprehensive waste management to ensure facilities are clean and safe, protect streams and groundwater from potential leaching and reduce unpleasant odors. There are some strategies to reduce the environmental impacts of horse excretions, such as using them as crop fertilizers or for the generation of renewable energy through anaerobic digestion. However, horse manure has not been considered viable for anaerobic digestion yet due to the high total solids content in combination with the presence of bedding materials with low process performance, such as wood chips and long straw which degrade slowly.

Scientific publications on the anaerobic digestion of excretions are scarce and mainly relate to pilot experiments or are performed at a laboratory scale [29]. In some regions of the US, straw-bedding horse manure is in high demand as fertilizer in mushroom farms [8]. There are also studies on methane gas ($CH_4$) emissions in chicken, turkey and horse excretions at different temperatures, and horse manure had a 13- to 130-fold increase in methane gas ($CH_4$) emissions.

According to what has been analyzed so far, the operational plan to be developed in the farms must be practical, carried out by a veterinary professional and consider the best

options regarding the use of water as well as excretion and bedding disposal depending on the availability of the place.

The benefits of the implementation of a comprehensive waste management plan in the farms mentioned before would be as follows:

- Healthier environment for horses;
- Cleaner and safer work areas;
- Protection of nearby water bodies and groundwater;
- Reduction of the volume of waste;
- Odor reduction.

Management measures should include the following:

- Location of bedding and excretion disposal facilities, mainly considering the distance to streams, ponds and water wells.
- Type and size thereof, depending on the amount of manure and the method of elimination, mainly considering that the floor is made of concrete, compacted clay or plastic to reduce the possibility of infiltration into groundwater. It can be designed using a waterproof base or a cover to keep it dry.
- Planning of elimination considering the rainy seasons.
- Use of covers to prevent manure storage and runoff (leaching) from manure piles entering streams and waterways.

## 4. Conclusions

The production of sport horses must be designed in such a way as to provide a consistent practical approach to environmental conservation. The characteristics of the different potential risk factors of this type of production (water use, feed, excretion management, location and distance from water bodies) must be considered to minimize environmental effects.

The assignation of the impact category of the farms allows for the implementation of concrete measures to minimize the environmental effects of this production. This methodology of characterization of the environmental impact can be used for other production facilities in various regions of the world.

**Author Contributions:** Conceptualization, M.M.V. and A.V.V.; methodology, M.M.V. and A.V.V.; sample collection, A.G.-L.; data curation, M.M.V. and A.G.-L.; software, M.M.V.; validation, M.M.V.; supervision, A.V.V. and A.F.-C.; writing—reviewing, M.M.V., A.V.V. and A.F.-C.; editing, M.M.V. and A.V.V.; project administration, A.F.-C.; funding acquisition, A.F.-C. and A.V.V. All authors have read and agreed to the published version of the manuscript.

**Funding:** This research was funded by CONICET, grant number P-UE 2018 INPA and Universidad de Buenos Aires UBACYT.

**Institutional Review Board Statement:** Not applicable.

**Informed Consent Statement:** Not applicable.

**Data Availability Statement:** http://repositoriouba.sisbi.uba.ar/gsdl/cgi-bin/library.cgi.

**Conflicts of Interest:** The authors declare no conflict of interest.

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
