# Peer review of "Environmental Effects of Sport Horse Production Farms in Argentina"

_sustainability, doi:10.3390/su151612210_

Round 1
Reviewer 1 Report
Dear authors!
Thank you for providing an interesting article regarding the consideration of the ecological state of horse breeding in Argentina. But in this manuscript we found several inaccuracies and corrections that we would like to get an answer to:
1. Line 1: Specify the type of article.
2. Line 14: The first sentence "The production of sport horses is expanding widely; " is better deleted.
3. Line 27-28: Replace "food" with "feed". It is advisable to add and specify keywords.
4. Line 45-48: Rewrite is not entirely clear to a wide audience. Pay attention to punctuation marks.
5. I would like the authors to take a broader look at the originality and significance of the work in the section "literature review". In this form, it boils down only to the fact that research is relevant only because of the lack of research, write in more detail. I would also like the authors to reduce the number of references in the literature review (because, in my opinion, there are very few of them for a small literature review).
6. Formulate the purpose of the work more concisely.
7. Table 1 is too wide, it seems that by doing this you increase the material. There are a lot of empty rows in front of the table.
8. Line 72-73 it is unclear what it is " AT: American trotter, B: Balance: N: Sport horse's production farms. PA: Polo Argentina, SA: The Power of Argentino. Z: Zangershaide"
9. Line 82-97: I think mentioning 1 time of table 2 is enough
10. The statement "So, for example, the composition of a balanced diet should be prescribed exclusively by veterinarians." The rations are not only veterinarians, but nutritionists and animal technicians.
11. The list of references is fulfilled according to the requirements of the journal.
Author Response
Buenos Aires, June 27, 2023
Dear Editor in Chief
I wish to submit an original research revised article entitled “Environmental effects of sport horses production farms in Argentina” by Mariana Vaccaro, Alejandra V.Volpedo, Alberto García Liñeiro and Alicia Fernández Cirelli for consideration by Sustainability. We thank the anonym reviewers for their suggestions which we have incorporated into the article. We attach the response at comments on the suggestions.
Please address all correspondence concerning this manuscript to avolpedo@fvet.uba.ar
Thank you for your consideration of this manuscript.
Sincerely,
Prof. Dra. Alejandra Volpedo
Response to Reviewer 1 Comments
We are grateful for the comments of reviewer 1
- Line 1: Specify the type of article.
Response 1: Specify the type of article was included.
- Line 14: The first sentence "The production of sport horses is expanding widely; " is better deleted.
Response 2: The sentence was deleted.
- Line 27-28: Replace "food" with "feed". It is advisable to add and specify keywords.
Response 3: The change was included.
- Line 45-48: Rewrite is not entirely clear to a wide audience. Pay attention to punctuation marks.
Response 4: The paragraph was modifícate.
- I would like the authors to take a broader look at the originality and significance of the work in the section "literature review". In this form, it boils down only to the fact that research is relevant only because of the lack of research, write in more detail. I would also like the authors to reduce the number of references in the literature review (because, in my opinion, there are very few of them for a small literature review).
Response 5: The literature was reviewed and the number of references was reduced, although suggestions from other reviewers had to incorporate some references
- Formulate the purpose of the work more concisely.
Response 6: The objective was written more concisely. 7. Table 1 is too wide, it seems that by doing this you increase the material. There are a lot of empty rows in front of the table.
Response 7: Table 1 was modificate.
- Line 72-73 it is unclear what it is " AT: American trotter, B: Balance: N: Sport horse's production farms. PA: Polo Argentina, SA: The Power of Argentino. Z: Zangershaide"
Response 8: The abbreviators were clarificated.
- Line 82-97: I think mentioning 1 time of table 2 is enough
Response 9: The suggestion was incorporated.
- The statement "So, for example, the composition of a balanced diet should be prescribed exclusively by veterinarians." The rations are not only veterinarians, but nutritionists and animal technicians.
Response 10: The suggestion was incorporated.

Reviewer 2 Report
The paper presents an analysis of the environmental effects of sport horse production farms in Argentina and proposes actions to minimize these effects. The study focuses on various characteristics such as proximity to water bodies, stall bedding practices, veterinary advice, feed composition, and water consumption. However, the analysis lacks quantitative data on the actual environmental impact of these farms. Additionally, the categorization of impact into low, medium, and high is subjective and would benefit from a more objective assessment. Further research and data collection are needed to provide a more comprehensive understanding of the environmental impact of sport horse production in Argentina and to validate the proposed recommendations.
The topic of analyzing the environmental effects of sport horse production farms in Argentina and proposing actions to minimize these effects can be considered relevant in the field of sustainable agriculture and animal production. It highlights the importance of considering environmental sustainability in equestrian sports, which is an area that may not receive as much attention as other aspects of horse breeding and training.
The paper addresses a specific gap in the field by focusing on sport horse production farms in Argentina and evaluating their environmental impact. While there may be existing research on the environmental effects of livestock farming in general, this study specifically targets the sport horse production sector, which may have its own unique characteristics and challenges.
By examining factors such as proximity to water bodies, stall bedding practices, and veterinary advice, the study aims to identify areas where improvements can be made to reduce the environmental impact of sport horse production. This specific focus on the sport horse industry fills a gap in the existing literature and provides insights that can be valuable for practitioners and policymakers in the field.
Compared to other published material, this paper on the environmental effects of sport horse production farms in Argentina adds several contributions to the subject area.
Firstly, it focuses specifically on the sport horse industry, which may have unique characteristics and management practices compared to other livestock farming sectors. This targeted approach allows for a more in-depth analysis of the environmental impact specific to sport horse production, which may differ from general livestock farming.
Secondly, the paper examines various factors such as proximity to water bodies, stall bedding practices, management practices, and veterinary advice in relation to feed composition and water consumption. By considering these specific variables, the study provides a comprehensive assessment of the environmental impact of sport horse production farms.
Additionally, the study presents a nominal qualitative analysis of the environmental impact, categorizing it into low, medium, and high impact. This classification provides a practical understanding of the severity of environmental effects associated with different farms, allowing for a more nuanced evaluation and comparison.
Furthermore, the paper highlights the role of professional veterinarians in minimizing the environmental impact of sport horse production. This emphasis on veterinary involvement adds a valuable perspective, recognizing the importance of expert guidance in sustainable management practices.
Overall, this paper contributes to the subject area by providing a focused analysis of the environmental effects specific to sport horse production farms, considering relevant variables and offering recommendations for sustainable practices. These insights fill a gap in the existing literature and contribute to the ongoing efforts to promote environmental sustainability in the sport horse industry.
line 31: The environmental effects of livestock productions have been studied worldwide in recent decades, i suggest citing: https://doi.org/10.1016/j.rvsc.2023.03.008
However, these are focused on cattle (better ruminants), i suggest add: 10.3390/ani13050797
and there are very few studies on horses and more specifically on sport horses and horses for meat production (citing:
10.1186/s12917-022-03289-2; 10.3390/ani12141740)
the introduction must be enlarged
horses characteristics must be reported
statistical analysis section is missing
In the discussion, it is important to consider additional factors such as forage quality, foreign bodies, and feed palatability. These factors can have a significant impact on the overall management and well-being of sport horses.
Forage quality plays a crucial role in meeting the nutritional requirements of horses and can affect their performance and health. It is important to assess the nutritional composition, digestibility, and potential contaminants in the forage. This aspect can be further explored by referring to the study with the citation 10.1016/j.jevs.2022.103940.
Furthermore, feed palatability is a critical aspect in ensuring that horses consume an adequate and balanced diet. Poor feed palatability can lead to reduced intake and potential nutrient deficiencies. The study 10.1016/j.applanim.2020.105110 provides insights into the assessment of feed palatability and its implications for equine nutrition.
By considering these additional factors and referencing the relevant studies, the discussion can be enriched with a more comprehensive understanding of the importance of forage quality, foreign bodies, and feed palatability in sport horse production.
Author Response
Buenos Aires, June 27, 2023
Dear Editor in Chief
I wish to submit an original research revised article entitled “Environmental effects of sport horses production farms in Argentina” by Mariana Vaccaro, Alejandra V.Volpedo, Alberto García Liñeiro and Alicia Fernández Cirelli for consideration by Sustainability. We thank the anonym reviewers for their suggestions which we have incorporated into the article. We attach the response at comments on the suggestions.
Please address all correspondence concerning this manuscript to avolpedo@fvet.uba.ar
Thank you for your consideration of this manuscript.
Sincerely,
Prof. Dra. Alejandra Volpedo
Response to Reviewer 2 Comments
We are grateful for the comments of reviewer 2.
The suggested references were incorporated. In addition, a paragraph was included in the discussion on forage quality.

Reviewer 3 Report
Dear Author, the study is interesting, to my opinion only a few corrections should be made, but some references are not pertinent with the text and must be adjusted.
Line 4: Mariana M. Vaccaro 1,2, Alejandra V Volpedo 3,4*, Alberto Garcia-Liñeiro1 and Alicia Fernández-Cirelli 3,4: what does 4 mean? What does it stand for?
Line 22: … analyzed is shown that..: has shown that
line 63: was administered on the: was administered to the
Table 1: in 17 and 21: correct: electrolytes instead of Electrolitos
Line 102: Reference n° 13 seems to me not pertinent with the text: it’s about “…removing heavy metals from industrial wastewaters…” not about overfeeding
Line 109: Reference n° 16 seems to me not pertinent with the text: it’s about the quality of hay, wrapped or not wrapped bales, molds etc. , not the use of trough
Line 144: in open-air deposits; intended…: in open-air deposits, intended…
Line 146 and following: Reference n° 19 seems to me not pertinent with the text at this point: it is about GHG emissions, there are not mentioned in that paper the mineral values reported in the text. Where do they come from? Moreover, I think that these values can vary widely, it doesn’t seem correct to me this statement: ” …excretions contain…” with precise values, not a range or an average, or did you measure these minerals by yourself (but this does not appear from the text)? In this case is it an average among the different farms?
Line 253:.. carry out extension activities for equine producers: … carry out further activities for equine producers
Line 254:.. sustainability in the of sport horses production farms: sustainability of sport horses production farms
Line 263: Reference n° 28: it is not findable, moreover in Swedish. It seems a slide show
Line 307: can be used in others productions: can be used in other productions
Line 368: phytate degradation: phytate degradation
Author Response
Buenos Aires, June 27, 2023
Dear Editor in Chief
I wish to submit an original research revised article entitled “Environmental effects of sport horses production farms in Argentina” by Mariana Vaccaro, Alejandra V.Volpedo, Alberto García Liñeiro and Alicia Fernández Cirelli for consideration by Sustainability. We thank the anonym reviewers for their suggestions which we have incorporated into the article. We attach the response at comments on the suggestions.
Please address all correspondence concerning this manuscript to avolpedo@fvet.uba.ar
Thank you for your consideration of this manuscript.
Sincerely,
Prof. Dra. Alejandra Volpedo
Response to Reviewer 3 Comments
We are grateful for the comments of reviewer 3.
1-Line 22: … analyzed is shown that..: has shown that
Response 1: The suggestion was incorporated.
2-line 63: was administered on the: was administered to the
Response 2: The suggestion was incorporated.
3-Table 1: in 17 and 21: correct: electrolytes instead of Electrolitos
Response 3: The suggestion was incorporated.
4- Line 102: Reference n° 13 seems to me not pertinent with the text: it’s about “…removing heavy metals from industrial wastewaters…” not about overfeeding
Response 4: The reference 13 was removed.
5-Line 109: Reference n° 16 seems to me not pertinent with the text: it’s about the quality of hay, wrapped or not wrapped bales, molds etc. , not the use of trough
Response 5: The reference 16 was removed.
6-Line 144: in open-air deposits; intended…: in open-air deposits, intended…
Response 6: The suggestion was incorporated.
7-Line 146 and following: Reference n° 19 seems to me not pertinent with the text at this point: it is about GHG emissions, there are not mentioned in that paper the mineral values reported in the text. Where do they come from? Moreover, I think that these values can vary widely, it doesn’t seem correct to me this statement: ” …excretions contain…” with precise values, not a range or an average, or did you measure these minerals by yourself (but this does not appear from the text)? In this case is it an average among the different farms?
Response 7: The suggestion was incorporated, and referente 19 was deleted.
8- Line 253:.. carry out extension activities for equine producers: … carry out further activities for equine producers
Response 8: The suggestion was incorporated.
9- Line 254:.. sustainability in the of sport horses production farms: sustainability of sport horses production farms
Response 9: The suggestion was incorporated.
10- Line 263: Reference n° 28: it is not findable, moreover in Swedish. It seems a slide show
Response 10: The reference 19 was removed.
11- Line 307: can be used in others productions: can be used in other productions
Response 11: The suggestion was incorporated.
12- Line 368: phytate degredation: phytate degradation
Response 12: The suggestion was incorporated.

Reviewer 4 Report
Dear Author,
Article is completely on survey based. There was no any parameters were studied to understand and corelate the role of environment on horse production.
I have gone through the complete manuscript, and after having a deep analysis, I am regret to say that article does not carry any relation/correlation on this.
authors are encouraged to submit again with changes in title or with new idea with the same survey.
Author Response
Buenos Aires, June 27, 2023
Dear Editor in Chief
I wish to submit an original research revised article entitled “Environmental effects of sport horses production farms in Argentina” by Mariana Vaccaro, Alejandra V.Volpedo, Alberto García Liñeiro and Alicia Fernández Cirelli for consideration by Sustainability. We thank the anonym reviewers for their suggestions which we have incorporated into the article. We attach the response at comments on the suggestions.
Please address all correspondence concerning this manuscript to avolpedo@fvet.uba.ar
Thank you for your consideration of this manuscript.
Sincerely,
Prof. Dra. Alejandra Volpedo
Response to Reviewer 4 Comments
We are grateful for the comments of reviewer 4.
The paper has been revised and clarifications incorporated. This work integrates previous work carried out by the study team (Vaccaro et al., 2020, 2021, 2022). In these works, the water quality (physical-chemical parameters and trace elements) of the 26 sport equine farms was determined; the presence of trace elements in food and dietary supplements, in excreta and in beds was analyzed. In addition, the water for direct use (drinking, animal bathing and box cleaning) and the virtual water contained in food, among other aspects, were quantified. With all these elements associated with the environmental impacts of the farms and the realization of a survey, this knowledge was integrated, a weighting was made and a scale of impact assessment was proposed that can be used in other farms. With all this, comprehensive management measures are suggested to reduce the negative impacts of this production.
Round 2
Reviewer 1 Report
Good afternoon!
Dear Authors!
Dear Editor!
The article can be accepted in its present form.
Reviewer 2 Report
After reviewing the revised version of the paper, I must say that it has significantly improved. The authors have addressed the issues raised during the review process and have made substantial revisions to enhance the quality of the manuscript. Therefore, I fully endorse the publication of this paper.
Best regards,
Reviewer 4 Report
Dear Author,
Thanks for your time and making changes,
After adding a paragraph and clarification, it is well under criteria for the analysis.
It is good.